# Characterization of the Kinetyx SI Wireless Pressure-Measuring Insole during Benchtop Testing and Running Gait

**DOI:** 10.3390/s23042352

**Published:** 2023-02-20

**Authors:** Samuel Blades, Matt Jensen, Trent Stellingwerff, Sandra Hundza, Marc Klimstra

**Affiliations:** 1School of Exercise Science, Physical & Health Education, University of Victoria, Victoria, BC V8W 2Y2, Canada; 2Canadian Sport Institute Pacific, Victoria, BC V9E 2C5, Canada

**Keywords:** plantar pressure, gait, running, smart insole, in-shoe, sensors, wearables

## Abstract

This study characterized the absolute pressure measurement error and reliability of a new fully integrated (Kinetyx, SI) plantar-pressure measurement system (PPMS) versus an industry-standard PPMS (F-Scan, Tekscan) during an established benchtop testing protocol as well as via a research-grade, instrumented treadmill (Bertec) during a running protocol. Benchtop testing results showed that both SI and F-Scan had strong positive linearity (Pearson’s correlation coefficient, PCC = 0.86–0.97, PCC = 0.87–0.92; RMSE = 15.96 ± 9.49) and mean root mean squared error RMSE (9.17 ± 2.02) compared to the F-Scan on a progressive loading step test. The SI and F-Scan had comparable results for linearity and hysteresis on a sinusoidal loading test (PCC = 0.92–0.99; 5.04 ± 1.41; PCC = 0.94–0.99; 6.15 ± 1.39, respectively). SI had less mean RMSE (6.19 ± 1.38) than the F-Scan (8.66 ±2.31) on the sinusoidal test and less absolute error (4.08 ± 3.26) than the F-Scan (16.38 ± 12.43) on a static test. Both the SI and F-Scan had near-perfect between-day reliability interclass correlation coefficient, ICC = 0.97–1.00) to the F-Scan (ICC = 0.96–1.00). During running, the SI pressure output had a near-perfect linearity and low RMSE compared to the force measurement from the Bertec treadmill. However, the SI pressure output had a mean hysteresis of 7.67% with a 28.47% maximum hysteresis, which may have implications for the accurate quantification of kinetic gait measures during running.

## 1. Introduction

Measurement of foot-ground interactions during running is critical to understanding and addressing the biomechanical factors related to overuse injuries [1,2,3] and improving performances [4,5]. Traditionally, standard laboratory-based measurement equipment such as in-ground force plates, or force-instrumented treadmills have been used to quantify foot-ground interactions during running. Increasingly, however, researchers have recognized the importance of measuring running foot-ground interactions outside of the laboratory in the natural training and competition environments [3,6]. Plantar pressure measurement systems (PPMS) offer a unique solution to researchers interested in measuring foot-ground interactions during running or walking, as well as for many different activities [7,8,9,10]. Unlike the standard laboratory tools mentioned previously, PPMS can be used both inside and outside of the laboratory, can continuously capture consecutive strides, and can record the distribution of pressures across the plantar surface of the foot [11,12]. Additionally, in-shoe PPMS have been shown to accurately capture important kinematic running gait metrics such as stance time, stride time, and stride rate [11,12,13,14,15,16]. Further, PPMS have shown promise to approximate kinetic characteristics of gait, such as path of center of pressure, and vertical ground reaction forces [6,10,14,17,18,19,20,21], although some research has shown inter-session force reliability is still inconsistent [22].

While PPMS show great potential for continuous running gait quantification, the design and form factor of most research-grade in-shoe PPMS, such as the industry-standard Tekscan F-scan system (Norwood, MA, USA) [22,23,24,25], are often bulky and can contain cumbersome components such as external cables, shoe-mounted pods, waist belts, and large external connectors Although designed for accuracy, the form factor of such research-grade systems have been shown to interfere with normal running gait [23], limiting their utility and potentially reducing the validity of their data [15]. Additionally, such systems require setup and supervision by trained researchers and are therefore not viable for data collection during many practical and unsupervised use cases. Finally, research-grade PPMS can be expensive, and are thus not viable for broad-scale consumer use. The development of a cost-effective PPMS with an unobtrusive and field-appropriate form factor has many important applications across research, clinical practice, and sports performance [3].

Recently, a new generation of PPMS with completely integrated pressure-measuring technology is emerging [26,27,28,29]. These pressure-enabled ‘smart insoles’ are fully contained and can be used by simply replacing the existing sock liner of the shoe with the smart insole. This provides a significant improvement in form factor and design over traditional research-grade PPMS [27]. By employing a fully integrated design, smart insoles are capable of unobtrusive gait measurement that can be used across a broad range of research and consumer use applications with minimal impact on the wearer’s natural gait. Additionally, due in part to their integrated form factor, smart insoles are able to be used broadly without researcher supervision [26,27]. One newly developed pressure-enabled smart insole is the Kinetyx Sensory Insole (SI) (Kinetyx Sciences, Calgary, AB, Canada, CAN; Figure 1). 

Similar to other smart insoles [18,26,29], the Kinetyx SI employs a fully integrated design, including an integrated printed circuit board (PCB) with measurement electronics, and a pressure-sensing layer with 32 discrete resistive pressure-sensing elements (Figure 1). Resistive pressure sensors are comprised of thin polymer films that exhibit a change in resistance with the application of pressure [30,31,32]. These sensors, with a thickness of only 0.2 mm, are widely used in alternate applications for their form factor, affordability, and versatility. Importantly, resistive pressure sensors are a more cost-effective method of measuring pressure over capacitive-style pressure sensors, which many of the research-grade PPMS employ. By employing resistive pressure sensors, smart insoles, such as the Kinetyx SI, become cost-effective, and thus hold tremendous potential for researchers and consumers by enabling broad-scale field appropriate use [30].

Despite its advantages, resistive pressure-sensing technology can vary widely in its accuracy and reliability [22,33,34]. Pressure sensor response characteristics such as drift, mean absolute error, and hysteresis can impact system accuracy both for pressure measurement and for derivative metrics such as ground reaction forces, center of pressure, and temporospatial gait metrics [17,22,34,35,36]. As such, resistive-based pressure-enabled smart insoles such as the Kinetyx SI should be fully validated against lab quality equipment to properly characterize their performance and accuracy before use in research or consumer applications.

To that end, Giacomozzi [34] established benchtop testing procedures to assess the performance of PPMS. For example, using this protocol, Giacomozzi was able to evaluate five different PPMS, including the Tekscan system, which was shown to be highly accurate relative to the other PPMS in benchtop testing. While this protocol has provided an important standardization of the methods used to characterize pressure-sensing technologies allowing for direct system-to-system comparisons [34], the loading rates of the sensors during standardized benchtop testing are far below those produced during running. Additionally, benchtop testing is executed without interaction with a human foot in situ and as such, may not properly capture the limitations of a given PPMS. For example, although the Tekscan system performed well in the assessment by Giacomozzi [34], research on the same system by Kati et al. [22] showed significant amounts of error in peak force measurement when used during sustained running. Thus, in order to fully characterize pressure-enabled smart insoles, the actual loading rates and conditions of running relative to gold standard gait biomechanics equipment is a necessary addition to any smart insole validation protocol [14,22,36]. Due to its improved form factor and fully integrated design, the SI system could offer an important new running sensor for researchers and clinicians, however, it has yet to be evaluated relative to an industry-standard PPMS during benchtop testing and research-grade biomechanical devices during athletic activities. If pressure-enabled smart insole systems can be determined as equivalent in response characteristics to laboratory-based systems, this can enable the potential to collect important mechanical data in situ.

Therefore, the purpose of this study was twofold. First, to assess the pressure sensor response characteristics of the Kinetyx SI system alongside a validated research-grade PPMS (Tekscan F-scan; Figure 2c) using an established PPMS benchtop validation protocol [34]. Second, to assess the pressure sensor response characteristics of the SI system during running across different speeds [14,37]. The results of this analysis will allow researchers and commercial users to understand the characteristics and limitations of the SI system and support ongoing evaluation and innovation of new smart insole and PPMS technologies.

## 2. Materials and Methods

### 2.1. Part 1—Benchtop Tests

#### 2.1.1. Benchtop Tests Data Collection

For the benchtop testing, the SI was tested alongside the sport version of the Tekscan F-Scan insert. Each PPMS was new at the time of testing and used according to the manufacturer’s specifications. The SI system comes pre-conditioned and calibrated; however, the F-Scan required conditioning, calibration, and equilibration before testing. A two-point calibration at 100 and 500 kPa was performed according to the manufacturer’s specifications on the sport version of the F-Scan system. The specifications for each of the PPMS are listed in Table 1.

Two testing devices, similar to those employed by Giacomozzi [34] were utilized to assess the PPMSs; a pneumatic bladder pressure tester (PBPT, Figure 2b) and a linear force testing device (LFTD; Figure 2a). The PBPT is a rigid structure with two parallel plates separated by a narrow gap where a PPMS insert is placed. An inflatable rubber membrane or bladder attached to the inside of the top plate can be inflated to apply uniform pressure across the entire surface of the PPMS. A digital pressure transducer (Greisinger Electronics GmbH, Regenstauf, Germany) continuously measured pressure within the bladder throughout each trial. The LFTD consisted of an arbour press and a force plate (AMTI, Watertown, MA, USA; Figure 2a). The force plate was secured to the base of the arbour press such that the arm of the arbour press was capable of applying an exact perpendicular load to the surface of the PPMS and the force plate simultaneously. All data was collected at 100 Hz using custom software (LabVIEW 2018 National Instruments, Austin, TX, USA). Four trials of each of the following four characterization tests were completed on two insoles from both the SI and F-Scan systems. All four trials of the benchtop tests were completed on the same day and were performed at standard room temperature and humidity. Results from the four trials were averaged for each test.

The tests were:

Step Test: The PBPT was used to apply 100 kPa steps of static pressure from 0 to 500 kPa and back to 0 kPa to the entire sensing area of the PPMS (Figure 2b). Each step had a minimum duration of 5 s and the PPMS were completely off-loaded for approximately 1 s after each step.

Sinusoidal Test: Using the LFTD, 10 sinusoidal pressure cycles ranging from 0–500 kPa were applied to each PPMS at approximately 1 Hz [34] (Figure 2a). The percent hysteresis was averaged over the central eight cycles of each trial.

Static test: Using the PBPT, constant pressure of 300 kPa was applied to the entire sensing area of the PPMS for 120 s [39]. The pressure gradient was measured as the maximum change in pressure over the central 40 s of the loading period.

Reliability Test: The SI and F-Scan inserts were loaded according to the step test protocol on three consecutive days and assessed for instrument test re-test reliability [39].

#### 2.1.2. Benchtop Tests Data Analysis

For the step test, the absolute and percentage root-mean-square error (RMSE), including their minimum, maximum, and mean values, along with Pearson correlation coefficient (PCC) were calculated. For the sinusoidal test, percent hysteresis, PCC and RMSE were calculated. For the static test, absolute error and the rate of error (slope) were calculated. For the reliability test, mean pressure values reported at each step from the step test were compared using Interclass correlation coefficients (ICC) (two ways mixed effects for absolute agreement). Direct comparisons (Student’s *t*-test) were made between SI and F-Scan values from the step test, sinusoidal test, and static test during the benchtop testing to determine if any statistical differences between the two PPMS existed on benchtop testing performance.

### 2.2. Part 2—Running

#### 2.2.1. Running Data Collection

To assess the overall responsiveness of the SI sensors during running, summed pressure data were concurrently collected from the SI sensors (200 Hz) while participants ran on a force-instrumented treadmill (1000 Hz, Bertec, Columbus, OH, USA, Figure 2d) [37]. All testing was conducted in the participants’ own athletic footwear, which were fitted with a correctly sized SI system after the existing shoe’s sock liners were removed. Each participant was instructed to perform 3 jumps at the start of each trial to create a manual sync event for post-hoc analysis. In total, 13 runners (8 male, 5 female) participants were recruited, aged 19–40 years (mean: 28 ± 5 years). Participant height ranged from 1.55 to 1.93 m (mean: 1.73 ± 0.10 m) and body mass ranged from 52.0 to 87.5 kg (mean: 66.6 ± 10.3 kg) [6]. All participants were free from injury at the time of testing, were familiar with treadmill running, and were given a 10 min self-selected warm-up. Each trial lasted for 60 s once the treadmill achieved steady state velocity. Each participant ran at 2.6, 3.0, 3.4 and 3.8 m/s with a self-selected rest interval between trials. These fixed speeds approximated a range of training and racing speeds used in previous studies of recreational runners [12,40,41]. Based on the results of previous research [22], the F-Scan system was deemed not suitable for running response evaluation. Thus, for this investigation only the SI sensors were assessed for responsiveness during running.

#### 2.2.2. Running Data Analysis

The vertical ground reaction force (vGRF) data and the SI pressure data were both resampled to 100 Hz. The Kinetyx SI pressure data was summed to create a sum of pressure signal (P_sum_) [37]. Once resampled, data from the force-instrumented treadmill and the P_sum_ from the SI system were synchronized post-hoc using cross-correlation. Both signals were normalized 0–100% based on the maximum values for each trial. To enable the running data analysis, the vGRF and P_sum_ signals needed to be broken into the loading and unloading phases of each step. To do this, the vGRF data were filtered using a zero-lag low-pass 10 Hz Butterworth filter for signal processing and signal peak detection [42]. A 5% threshold was then used to determine signal onset and offset for each loading cycle [43]. Next, a peak detection algorithm was used on the filtered signals to determine the local maximum of each loading cycle (see Figure 3).

For each trial, a mean loading and mean unloading trace was developed (see Figure 4). These mean signals were generated per trial and used to generate the following statistical measures. To assess the agreement between the SI and vGRF signal, Pearson’s correlation coefficient (PCC), root mean squared error (RMSE), and percent hysteresis were computed per trial, similar to the sinusoidal trial in part 1 of this investigation. Repeated measured ANOVA were executed across speeds for each of the statistical measures.

## 3. Results

### 3.1. Results Part 1

The results from the step, static, sinusoidal, and reliability tests for the SI and F-Scan systems are summarized in Table 2. The SI had similar linearity (Pearson’s correlation coefficient, PCC = 0.86–0.97) and mean RMSE (9.17 ± 2.02) compared to the F-Scan (PCC = 0.87–0.92; RMSE = 15.96 ± 9.49) on the step test. The SI and F-Scan had comparable results for linearity and hysteresis on the sinusoidal test (PCC = 0.92–0.99; 5.04 ± 1.41) (PCC= 0.94–0.99; 6.15 ± 1.39), respectively. SI had statistically less mean RMSE (6.19 ± 1.38) than the F-Scan (8.66 ± 2.31) on the sinusoidal test and statistically less absolute error and slope (4.08 ± 3.26, 0.10 ± 0.08) than the F-Scan (16.38 ± 12.43, 0.41 ± 0.31) on the static test. The SI had comparable measures of reliability (interclass correlation coefficient, ICC = 0.97–1.00) to the F-Scan (ICC = 0.96–1.00).

### 3.2. Results Part 2

The results from the running tests are summarized in Table 3. In total, 57 trials were analyzed for a total of 4031 foot-ground interactions. The SI pressure output had a near-perfect linearity and low RMSE compared to the vGRF signal. The SI pressure output also had a mean hysteresis of 7.67% with a 28.47% maximum hysteresis.

## 4. Discussion

Overall, this study has demonstrated that the SI system displayed a high level of accuracy and reliability when compared to an industry-standard PPMS during standard benchtop testing. Additionally, when compared to a force-instrumented treadmill during running, the SI system pressure output showed strong correlation to the force measurement. However, during running there was notable hysteresis that may impact the generation of force-related gait metrics. Taken together, these results support the use of this smart PPMS insole as a valid and reliable tool for field-based running assessment.

For the benchtop testing in this study, the methodology employed was chosen based on a previous study by Giacomozzi et al. [34]. During the series of benchtop tests, the SI system showed comparable results to the F-Scan during the step test and showed significantly less sinusoidal mean RMSE and static load error when compared to the F-Scan system. Further, both systems had near-perfect between-day reliability. It is important to note that there are differences between our results for the F-Scan system and those generated by Giacomozzi et al. [34]. In this study the linearity of the F-Scan system was lower than reported by Giacomozzi et al. [34]. During the step test and static test, the mean RMSE and error were similar but varied substantially between F-Scan sensors 1 and 2. During the sinusoidal test, the correlation and hysteresis were similar between the current study and Giacomozzi et al. [34], while the RMSE was much lower in this study. Some of the differences in the benchtop tests may be due to variability in sensors from the same manufacturer as well as potential differences in the measurement hardware between studies. While this study did not employ the exact testing equipment as Giacomozzi et al. [34] the PBPT was designed specifically for PPMS testing and the linear force testing used a research-grade force plate and a precision linear press. The results of the day-to-day reliability for both sensors in the present study, with high to near-perfect ICCs, would suggest that the variability of the specific sensors from the same manufacturer might be the cause for discrepancy between results of this study and that of Giacomozzi et al. [34] and not the testing equipment. The replication of most findings shows the repeatability and importance of conducting benchtop testing to evaluate PPMS based on the standards set out by Giacomozzi et al. [34]. While the results demonstrate comparable characterization of the SI system to an industry-standard PPMS during benchtop tests, it is also important to understand the response of PPMS across a range of tasks such as running.

Although benchtop testing is important to characterize and compare pressure measurement systems, the testing parameters may be insufficient to assess PPMS that are to be used in highly dynamic loading activities such as running. The linearity of the SI insole compared to the force measurement was near-perfect, which is expected as pressure is the force divided by the area perpendicular to the applied force. This covariance supports the development of similar spatiotemporal gait metrics such as stride and stance timing and rate between pressure and force output [11,13,14,33]. However, the hysteresis findings, as seen across subjects in Table 3, demonstrate that there are differences in loading and unloading responses between the SI sensors and the force-instrumented treadmill. These differences in loading rates and the sensor responses are important to consider, as rate of force and pressure development during loading and unloading can impact the estimation of kinetic and kinematic metrics during gait. The loading rates during running are much higher than the loading rates of the benchtop sinusoidal test. This demonstrates the benefit of in situ testing such as running to properly characterize PPMS. Further, these results support the investigation of algorithms to transform SI pressure measurement to force output. While measures of peak ground reaction force could be established without complex methods, the ability of the SI sensor to display valid force outputs on a full range of ascending and descending values will require focused techniques to ensure accurate values. Future investigations will be important to establish the SI sensor against other PPMS such as the Novel Loadsol that has demonstrated strong agreement against the Bertec instrumented treadmill [14]. Further, in their study, Burns et al. [14] showed strong agreement between the Loadsol sensor and force measurement across different tasks. Comparisons such as those presented by Burns et al. [14] have great value for evaluating PPMS, and the SI sensor should be put through similar evaluations before being used in different tasks.

A limitation of this study design is that the PPMS being assessed are not being tested simultaneously in situ. For example, where multiple sensors are placed in the same shoe and measuring the same foot impact. While such tests would provide direct comparisons between systems, previous research has shown that there is an interference effect between in-shoe pressure insoles systems where the order in which they are placed within the shoe alters the pressure measurement of each system [17]. Another important limitation of this characterization is that the testing was conducted under constant (benchtop) or unknown (running) conditions of heat and humidity which have also been shown to affect resistive pressure sensors as are used in the SI system [5]. Further testing of the SI system under ranges of these conditions is warranted.

A limitation of the SI system is that it utilizes 32 discrete pressure-sensing elements and thus does not have complete coverage of all points of pressure application. In contrast, the F-Scan system utilizes a continuous array of 960 sensors and thus can capture all points of pressure application under the foot. The lack of complete coverage has implications for the calculation of contact area, which was not measured in this study [8]. Additionally, the lack of comparable spatial resolution of the SI system could have implications for subsequent biomechanical metric calculations, such as vertical ground reaction force estimation [3] and center of pressure, particularity at the extreme boundaries of the foot where coverage is potentially limited. Despite these limitations, the overall form factor of the SI system has several advantages. At 65 g per insole, it is substantially lighter than the F-Scan system at 862 g. By simply replacing the running shoe’s original sock liner, the SI system can potentially be worn without interfering with natural running gait. The F-Scan however, requires the participant to wear measurement hardware on their waist, including cables and large connectors mounted to the lateral aspects of their legs connecting the various components. PPMS with this form factor has been shown to interfere with natural running gait [23], potentially limiting the validity of gait measurements made with the F-Scan system. It is important to note that at the time of testing, the most recent version of the F-Scan system was not available. It is possible that recent improvement in F-Scan sensor characteristics may support greater accessibility of this manufactured sensor for running-based assessment.

## 5. Conclusions

Overall, the SI PPMS performed similarly to or better than the F-Scan system during benchtop testing. Specifically, the SI displayed less mean RMSE during sinusoidal loading and less absolute error during static loading. Both the SI and F-Scan had near-perfect between-day reliability. During the running assessment, the SI pressure output had a near-perfect linearity and low RMSE compared to the force measurement from the Bertec treadmill. However, the SI pressure output displayed a broad range of hysteresis which may have implications for the accurate quantification of kinetic gait measures during running.

These results highlight the need for future research to support further characterization of smart insoles such as the Kinetyx SI during different athletic tasks. Additionally, this research highlights the need for an expanded standardized testing protocol that can be employed for the testing of smart insoles being used in dynamic activities such as running. Finally, this research evaluated the potential for resistive-based pressure measurement technology to provide a cost-effective alternative for use in smart insoles. Further development of smart insoles such as the SI may enable in situ running gait data collections to address biomechanical factors related to running related injuries and improve performances [3].

## Figures and Tables

**Figure 1 sensors-23-02352-f001:**
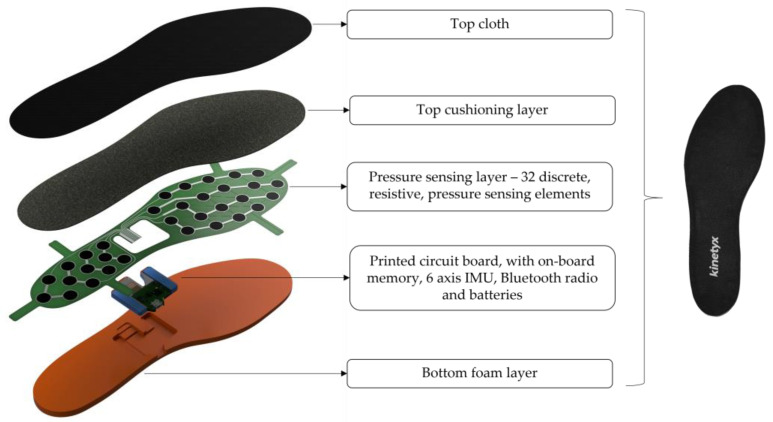
Expanded view of the Kinetyx SI System displaying the main components of the system contained within the fully integrated design, including a pressure-sensing layer (green) which contains 32 resistive pressure-sensing elements distributed across the rearfoot and forefoot regions.

**Figure 2 sensors-23-02352-f002:**
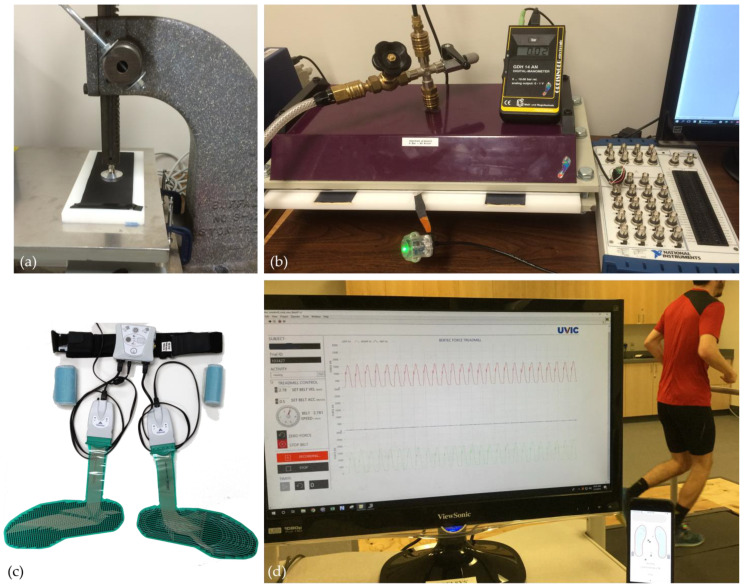
Part 1 testing equipment: (**a**) linear force testing device with 30 mm diameter actuator disk and force plate used for the sinusoidal testing (**b**) pneumatic bladder pressure tester used for step test, static test, and reliability test (**c**) Tekscan F-scan system. Part 2 testing equipment: (**d**) Bertec force-instrumented treadmill.

**Figure 3 sensors-23-02352-f003:**
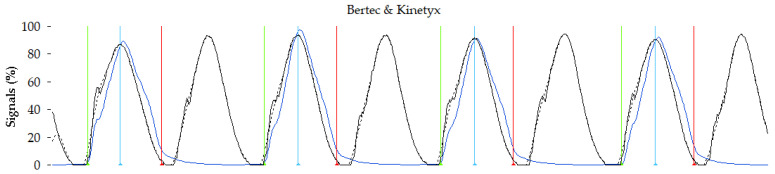
Normalized vGRF signal from the Bertec force-instrumented treadmill (black) and the filtered version of this signal (black dashed). Normalized pressure sum signal from the Kinetyx SI system (blue). Detection of signal onset (green), signal max (blue) and signal offset (red) for each loading cycle from a given foot from the Bertec vGRF data.

**Figure 4 sensors-23-02352-f004:**
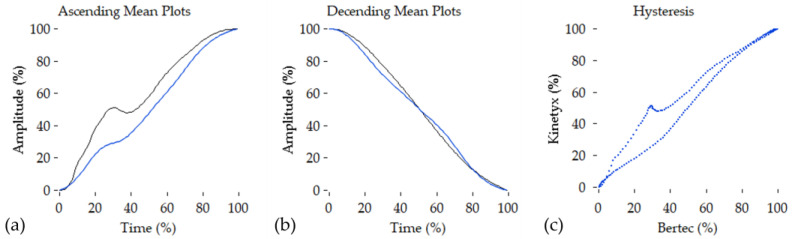
Mean loading plots from the running data. (**a**) shows the mean normalized loading data from the ascending part of the signal from the Bertec (black) and SI (blue); (**b**) shows the mean normalized loading data from the descending part of the signal from the Bertec (black) and SI (blue); (**c**) shows the mean hysteresis plot of SI vs. Bertec.

**Table 1 sensors-23-02352-t001:** Properties and listed specifications for each system. Information on the Tekscan system was taken from the Tekscan F-Scan data sheet [38]. Information on the Kinetyx SI system was supplied by the manufacturer upon request.

Plantar Pressure Measurement System Characteristics		
	Kinetyx SI	Tekscan F-Scan
Technology	Resistive	Resistive
System weight—both sides (g)	130	862
Number of sensing elements	32	960
Max sample rate (Hz)	200	750
Resolution	12 bit	8 bit
Sensing range (kPa)	0–500	0–862
Insert Thickness (mm)	5 mm	0.2
Durability (uses)	Unknown	5–15

**Table 2 sensors-23-02352-t002:** Results from the benchtop testing of the Kinetyx SI and the Tekscan F-Scan system. Results are from tests evaluating both the absolute accuracy of the systems (Step Test, Sinusoidal Test) and for sensor response characteristics (Sinusoidal Test, Static Test) and for reliability (Test -ReTest Reliability). Significant differences between systems are shown in bold.

	Kinetyx			Tekscan		
PPMS	SI 1	SI 2	SI Avg.	F-Scan 1	F-Scan 2	F-Scan Avg.
Step Test (0–500 kPa)						
Linearity R^2^ (range)	0.87–0.97	0.86–0.90	0.86–0.97	0.87–0.91	0.90–0.92	0.87–0.92
Mean RMSE (kPa)	10.72 ± 1.72	7.62 ± 0.32	9.17 ± 2.02	8.91 ± 4.47	23.01 ± 7.59	15.96 ± 9.49
MAX RMSE (kPa)	33.11 ± 5.61	19.85 ± 1.72	26.48 ± 8.06	17.66 ± 8.96	40.87 ± 13.04	29.27 ± 16.16
MIN RMSE (kPa)	1.85 ± 0.40	2.27 ± 0.82	2.06 ± 0.64	1.06 ± 0.39	0.69 ± 0.14	0.88 ± 0.34
Sinusoidal Test (0–500 kPa load cycling at ~1 Hz)						
Correlation R^2^ (range)	0.98–0.99	0.92–0.98	0.92–0.99	0.97–0.99	0.94–0.99	0.94–0.99
RMSE (kPa)	5.33 ± 0.53	7.05 ± 1.49	**6.19 ± 1.38**	10.47 ± 1.74	6.86 ± 0.87	**8.66 ± 2.31**
Hysteresis (%)	5.34 ± 1.57	4.75 ± 1.39	5.04 ± 1.41	6.85 ± 1.45	5.45 ± 1.04	6.15 ± 1.39
Static Test (The central 40 sec. of a 120 sec. window held at 300 kPa)						
Total Error (kPa)	3.80 ± 4.34	4.37 ± 2.38	**4.08 ± 3.26**	26.98 ± 7.64	5.78 ± 1.63	**16.38 ± 12.43**
Slope (kPa/sec)	0.09 ± 0.11	0.11 ± 0.06	**0.10 ± 0.08**	0.67 ± 0.19	0.14 ± 0.04	**0.41 ± 0.31**
Test Re-Test Reliability (ICCs)						
ICC	0.995	0.998		0.988	0.997	
ICC (95%) (Lower)	0.973	0.989		0.669	0.957	
ICC (95%) (Upper)	0.999	1.000		0.999	1.000	
Day to Day Variability	10.02 ± 4.88	7.66 ± 2.95		16.92 ± 3.75	7.9 ± 3.32	

**Table 3 sensors-23-02352-t003:** Running testing results by subject, for the Kinetyx SI as compared to the vGRF data from the force-instrumented treadmill.

Participants	1	2	3	4	5	6	7	8	9	10	11	12	13
2.6 (m/s)													
RMSE (%)	11.9	6.7	15.0	8.6	5.1	13.0	7.9	12.2	9.0	9.8	11.1	9.7	9.0
R^2^	0.97	1.00	0.96	0.99	0.99	0.98	0.99	0.96	0.97	0.99	0.98	0.99	0.99
Max Hysteresis (%)	26.7	15.6	44.2	15.8	11.3	35.9	21.2	35.5	20.6	25.6	32.8	19.8	31.8
Mean Hysteresis (%)	8.6	5.2	11.3	6.9	3.7	9.4	5.8	8.9	6.7	6.7	7.9	7.5	6.0
3.0 m/s													
RMSE (%)	12.9	7.6	13.6	9.3	7.5	13.1	7.5	14.1	8.6	10.3	11.7	10.3	9.1
R^2^	0.96	0.99	0.95	0.99	0.99	0.98	0.99	0.91	0.97	0.98	0.98	0.98	0.99
Max Hysteresis (%)	30.0	17.5	45.7	17.7	16.7	35.7	20.5	43.2	19.6	27.1	37.6	23.5	33.0
Mean Hysteresis (%)	9.2	5.8	9.8	7.4	6.0	9.6	5.5	9.6	6.4	7.5	8.0	7.7	6.2
3.4 (m/s)													
RMSE (%)	12.4	13.7	9.4	7.0	13.8	7.1	15.0	8.6	8.9	10.6	12.4	9.8	12.4
R^2^	0.96	0.95	0.99	0.99	0.97	0.98	0.85	0.95	0.97	0.97	0.99	0.98	0.95
Max Hysteresis (%)	30.5	42.2	20.4	17.2	36.3	17.8	44.9	22.7	26.5	34.8	24.9	34.8	32.9
Mean Hysteresis (%)	8.6	9.9	7.0	5.3	10.0	5.5	9.9	6.3	6.1	6.5	9.6	6.8	8.6
3.8 (m/s)													
RMSE (%)	13.3	13.7	8.4	6.9	15.1	6.5	14.6	8.5	12.6	11.8	11.2	12.4	10.8
R^2^	0.95	0.94	0.99	0.99	0.96	0.97	0.75	0.95	0.94	0.99	0.97	0.96	0.98
Max Hysteresis (%)	32.6	42.6	19.5	18.8	41.3	21.1	42.9	24.6	39.3	25.5	40.6	30.0	27.0
Mean Hysteresis (%)	9.3	10.0	6.3	5.0	10.9	4.7	10.5	6.1	8.0	8.7	7.6	8.9	8.5

## Data Availability

Not applicable.

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
