# Peer review of "Characterization of the Kinetyx SI Wireless Pressure-Measuring Insole during Benchtop Testing and Running Gait"

_sensors, 2023, doi:10.3390/s23042352_

Round 1

Reviewer 1 Report

In this work, for Characterization of the Kinetyx SI wireless pressure-measuring 2 insole during benchtop testing and running gait. In my opinion, this work has good scientific significance and application value. Therefore, this manuscript can be considered for publication after addressing the following issues:

1.      In the Introduction part, a total of 10 literatures were cited to describe the background of this work. However, only 1 literature were from 2019 and beyond. This does not sufficiently present new research results in the field and does not illustrate the novelty of this work.

2.      Please explain the Conclusions more clearly and defined some accuracy values.

3.      Author should explain the Compete system block diagram as well as Complete setup comprising of the insole, 3D printed box and the printed circuit board,

4.      Author should show figure 1 in colored, why author showed in black and white in this modern era. Author must change all over figure all section figure 1e,f,g not shown clearly.

5.      There are a few typos in the reference which need to correct carefully. For example- “C.Giacomozzi, “Hardware performance assessment recommendations and tools for baropodometric sensor systems,” p. 10” please add Year,

In this reference “H. L. P. Hurkmans, J. B. J. Bussmann, E. Benda, J. A. N. Verhaar, and H. J. Stam, “Accuracy and repeatability of the Pedar 376 Mobile system in long-term vertical force measurements,” p. 8, 2006.” Dio added is left, please author go through carefully all over the manuscript.  

Reviewer 2 Report

Please substantially strengthen the statement on the theoretical basis and algorithm of the research in the paper, and further elaborate the specific significance and value of the research content.

Reviewer 3 Report

In this work, the authors carried out a benchtop characterization of the Kinetyx SI system alongside a validated PPMS industry standard using established PPMS validation protocols and methodologies, and compared the SI system to a Bertec instrumented treadmill to assess the rate of sensor response during running. The content of this manuscript is clear and logical, and the results of this work may allow researchers and commercial users to understand the characteristics and imitations of the SI system and support ongoing evaluation and innovation of new PPMS technology. However, there is still some problems, a minor revision is needed:

1.     In Table 1, the durability of the Tekscan F-Scan is 5 – 15*, I’m not sure how much that means. For important parameters, I think authors can provide more clear quantitative methods for readers to understand, or add more typical work for comparison, so that readers can easily know the current progress of key parameters and understand the value of performance comparison and analysis in the article. Besides, the whole manuscript needs to be carefully checked to ensure that it conforms to the format requirements of the journal. For example, the semicolon format of the "Sensing range" in Table 1 is not uniform, which will affect the reading.

2.     Figure 1 provides pictures of test objects and equipment. However, to better demonstrate the research method and facilitate readers to reproduce experiments, I think authors can provide pictures to introduce the test scenes.

Reviewer 4 Report

The manuscript by Blades et al. evaluated pressure measurement error and reliability of a new commercially available plantar-pressure measurement system (PPMS) versus an industry-standard PPMS under different conditions. This work adds to the field as the use of newer generation PPMS has important applications across research, clinical practice, and sports performance. Overall, I have no major concerns regarding the manuscript. However, there are some minor points that would help to improve the manuscript.

1) Page 5, line 175: please explain “synchronized post-hoc”. How, exactly?

2) I assume that RMSE (e.g., “RMSE=15.96 ± 9.49”) is reported as a percentage (e.g., “RMSE=15.96% ± 9.49%”)?

3) I was not able to find the details regarding the ICC. For example, which ICC model was used? Does this represent consistency or absolute? Single or average measures?

4) More details regarding the statistical analysis are needed. For example, how was the (paired?) "Student’s T-test" applied in the study?  

Round 2

Reviewer 2 Report

Please strengthen the theoretical value and application prospect of the research in the paper.
